# Molecular and Hormonal Aspects of Drought-Triggered Flower Shedding in Yellow Lupine

**DOI:** 10.3390/ijms20153731

**Published:** 2019-07-31

**Authors:** Emilia Wilmowicz, Agata Kućko, Sebastian Burchardt, Tomasz Przywieczerski

**Affiliations:** 1Chair of Plant Physiology and Biotechnology, Nicolaus Copernicus University, 1 Lwowska Street, 87-100 Toruń, Poland; 2Department of Plant Physiology Warsaw, University of Life Sciences-SGGW (WULS-SGGW), Nowoursynowska 159 Street, 02-776 Warsaw, Poland

**Keywords:** abscisic acid, abscission zone, drought stress, ethylene, hormone homeostasis, yellow lupine, *LlIDA*, *LlHSL*, *LlMPK6*, catalase

## Abstract

The drought is a crucial environmental factor that determines yielding of many crop species, e.g., *Fabaceae*, which are a source of valuable proteins for food and feed. Herein, we focused on the events accompanying drought-induced activation of flower abscission zone (AZ)—the structure responsible for flower detachment and, consequently, determining seed production in *Lupinus luteus*. Therefore, detection of molecular markers regulating this process is an excellent tool in the development of improved drought-resistant cultivars to minimize yield loss. We applied physiological, molecular, biochemical, immunocytochemical, and chromatography methods for a comprehensive examination of changes evoked by drought in the AZ cells. This factor led to significant cellular changes and activated AZ, which consequently increased the flower abortion rate. Simultaneously, drought caused an accumulation of mRNA of genes *inflorescence deficient in abscission-like* (*LlIDL*), *receptor-like protein kinase HSL* (*LlHSL*), and *mitogen-activated protein kinase6* (*LlMPK6*), encoding succeeding elements of AZ activation pathway. The content of hydrogen peroxide (H_2_O_2_), catalase activity, and localization significantly changed which confirmed the appearance of stressful conditions and indicated modifications in the redox balance. Loss of water enhanced transcriptional activity of the abscisic acid (ABA) and ethylene (ET) biosynthesis pathways, which was manifested by elevated expression of *zeaxanthin epoxidase* (*LlZEP*), *aminocyclopropane-1-carboxylic acid synthase* (*LlACS*), and *aminocyclopropane-1-carboxylic acid oxidase* (*LlACO*) genes. Accordingly, both ABA and ET precursors were highly abundant in AZ cells. Our study provides information about several new potential markers of early response on water loss, which can help to elucidate the mechanisms that control plant response to drought, and gives a useful basis for breeders and agronomists to enhance tolerance of crops against the stress.

## 1. Introduction

Global climate changes have a significant effect on the temperatures on Earth, making the planet increasingly warmer. Nowadays, the resulting soil drought is one of the most important factors limiting plant growth and productivity of many crop species. It is estimated that in 25 years, this negative effect of water loss will result in yield reduction by up to 30% [1]. The *Fabaceae* family species, among them yellow lupine (*Lupinus luteus* L.), is affected by the strong correlation between weather conditions and yielding. Since lupine seeds are characterized not only by high protein content, but also by having good nutritional properties and a unique and desirable protein composition, this species is a sustainable food and feed source. Additionally, studies performed in recent years have indicated that its consumption provides numerous health benefits; for example, lupine ingredients have started to be used in probiotic and nutraceutical production. The potential value of lupine in diabetes and obesity management may be of relevance to public health. Furthermore, the plant’s ability to fix nitrogen, and the resulting possibility of supplying that element to the following crops, are of interest to agronomists [2,3]. It is of extreme importance to get the knowledge concerning the physiological aspects and mechanisms of the plant’s response to drought stress, which includes identifying the molecules responsible for its adaptation and survival under unfavorable conditions. 

Plant production is highly dependent on proper generative organ formation and detachment being ensured until the seed formation and filling stages. Drought stress leads to excessive and premature flower abscission, which blocks pod development and has a strong limiting effect on yielding. The place of flower detachment is named the abscission zone (AZ). It is a specialized, spatially-limited structure sensitive to unfavorable conditions [4]. Therefore, understanding how AZ activity is regulated is of supreme importance to control yielding.

According to our previous studies, the function of the AZ in yellow lupine depends on the activity of phytohormones, such as ethylene (ET), abscisic acid (ABA) and auxin, which interact with one another and are contributed in the subsequent AZ activation stages [5,6]. We have shown that ABA influences the ET biosynthesis pathway, which is related to the *aminocyclopropane-1-carboxylic acid synthase* (*LlACS*) gene, the ET precursor aminocyclopropane-1-carboxylic acid (ACC), and the *aminocyclopropane-1-carboxylic acid oxidase* (*LlACO*) gene. The AZ exhibits specific features characteristic for activation processes, which are also observed in the cellular ultrastructure [5]. We also revealed that these modifications are accompanied by significant changes in the mRNA content and tissue localization of the *blade on petiole* (*LlBOP*) transcript [7,8]. The studies performed in recent years on *Arabidopsis thaliana*, *Litchi chinensis,* and *Solanum lycopersicum* indicated the presence of numerous genetic components coordinating events that take place in the AZ cells, e.g., *inflorescence deficient in abscission* (*IDA*), *haesa*/*haesa-like2* (*HAE*/*HSL2*), *nevershed* (*NEV*), and *mitogen-activated protein kinase* (*MKK4*/*MKK5 and MPK6*). IDA is a ligand for leucine-rich repeat receptor-like kinases HAE/HSL2. The formation of the IDA–HAEHSL2 complex induces a MAP cascade within the cytoplasm of AZ cells that activates transcriptional factors inducing organ abscission [9,10]. Our findings showed that a homolog of the *IDA* gene in yellow lupine (*LlIDA*) is upregulated by ABA and ET, suggesting cross-talk among phytohormonal and genetic pathways regulating flower separation, which is an intriguing and unexplored area of research [11]. Recently, we have also presented specific localization of reactive oxygen species in the AZ area following its activation, and paid attention to processes related to oxidative stress [8]. 

Despite considerable advancements in the research of abscission processes, still little is known about the mechanism activated in AZ cells during environmentally-induced abscission, such as in drought. Therefore, this study aims to analyze the contribution of various elements of hormonal and genetic pathways of flower abscission in AZ-specific response to water deficit. In the presented paper, the positive correlation between water deficit and cellular specific changes in AZ cells was explored. We found that subsequent elements of the genetic pathways that activate abscission (*LlIDA*, *LlHAE*, *LlMPK6*) were triggered in response to drought. Additionally, we observed disruption in the redox balance. Our findings give a novel insight into the interplay between drought and abscission. Finally, we propose a molecular model of AZ cells’ functioning under drought stress conditions.

## 2. Results

### 2.1. Lupine Biometric Parameters and Elements Composition Are Affected by Drought

Lupines cultivated under drought developed fewer leaves when compared to plants subjected to optimal moisture conditions (Appendix A). The average leaf area in the stress-treated plants was lower (~33%) than in the control (Appendix A). Additionally, drought decreased their photosynthetic activity. The Fv/Fm parameter reached lower values in the leaves of drought-treated lupines (Appendix A). In the next step, we found that plant cultivation under soil drought conditions decreased leaf water content (Appendix A). Drought-stressed plants exhibited a different element composition in the leaves. Fe content dropped, both on the 48th and 51st days of cultivation (Appendix A). It should be noted that the content of this microelement, in the stressed plants and in the control, was significantly lower on the 48th day when compared to 51st. The drought decreased the concentrations of K and P, as well, and this effect was age-dependent (Appendix A). The younger leaves of the control plants and the drought-stressed organs contained more K than the older ones (Appendix A). A similar tendency was observed for P (Appendix A). Soil drought-stress caused significant differences in the Na, S, and Zn contents in the leaves of 51-day-old plants (Appendix A), whereas it had no effect on their level in 48-day-old plants, except for Na (Appendix A). The largest differences in the drought-treated plants when compared to the control were observed for Na. 

Drought stress in the studied AZ fragments was manifested by lower water content (Figure 1A) and had a positive effect on the aborted flower percentage (Figure 1B). The control plants cultivated under optimal soil moisture conditions developed an average of 12 flowers, with 60% of them becoming aborted. Soil drought increased this parameter’s value, causing ~90% of the flowers to be aborted (Figure 1B). 

### 2.2. AZ Cells Exhibited Different Structure During Drought

Drought caused middle lamellae dissolution, which consequently led to cell-to-cell adhesion loss and tissue integrity disruption in the AZ area (Figure 2E,F). The cells contained many cellular aggregates and large protein-enriched nuclei (Figure 2G,H). Coomassie staining also revealed higher protein contents in the cells after drought treatment (Figure 2G,H) than in the control (Figure 2C,D). Inactive AZ cells were round and loosely arranged (Figure 2A,B). 

### 2.3. Drought Induces Expression of Components of IDA/HAE/MPK6 Pathway in AZ 

A complete cDNA sequence of the *LlHSL* gene was identified, and our analyses of the predicted LlHSL protein performed using ProtComp indicated its membrane localization (Appendix A). By sequencing PCR products, a cDNA fragment of the *LlMPK6* gene was obtained (Appendix A). 

Drought significantly modulated the transcriptional activity of the gene encoding elements of the molecular abscission-associated pathway: *LlIDL*, *LlHSL*, and *LlMPK6* (Figure 3). *LlIDL* expression was five times higher in the floral AZs of the stressed plants as compared with the control on the 48th day of cultivation and maintained high in the subsequent stage (Figure 3A). *LlHSL* transcription was also significantly up-regulated in the subsequent time-variants of the plants subjected to drought (Figure 3B). It should also be pointed out that the expression of *LlHSL*, similarly to *LlMPK6*, was enhanced in the floral AZs of non-stressed plants during development (48th and 51st days, Figure 3B,C). Nevertheless, the strong stimulation of the *LlMPK6* gene was noticed in the floral AZs of the plants subjected to drought, both on the 48th and 51st days of their development (Figure 3C).

In the floral AZ cells of the plants cultivated under soil drought conditions, a high level of fluorescence, indicating the presence of MPK6, was observed (Figure 4C). What is more, a strong signal was noted in the vascular bundle (Figure 4D,E). Fluorescence was detected within the cytoplasm, which contained numerous aggregates emitting an intensive signal (Figure 4E). On the other hand, the non-active AZ cells contained much less MPK6 (Figure 4B). 

### 2.4. Water Deficit Influences H_2_O_2_ Level and Changes Localization and Activity of Catalase 

Drought stress caused an increase in the level of hydrogen peroxide in the flower AZ (Figure 5H) and enhanced the activity of catalase (CAT) (Figure 5G). Additionally, a strong fluorescent signal, indicating the presence of CAT, was found in the drought-stressed AZ cells (Figure 5C,E) and the cells adjacent to the vascular bundles (Figure 5D). Soil moisture had a significant effect on the diversified CAT localization in the vascular bundles, as well. The inside of the vascular bundle cells from the drought-stressed floral AZs was completely filled with CAT (Figure 5F), while traces of fluorescence were observed in the control AZ (Figure 5A) and the nearby vascular bundles (Figure 5B). 

### 2.5. Correlation Between Stress Phytohormones and Drought

A complete cDNA sequence of the *LlZEP* gene was obtained (Appendix A). In the plants cultivated under optimal conditions, the *LlZEP* expression and endogenous ABA levels increased during lupine development (Figure 6A,B). Furthermore, drought stress had a positive influence on the transcriptional activity of the ABA biosynthesis gene and the phytohormone content (Figure 6A,B). Their maximum values were almost three times higher in comparison to the control plants (Figure 6). The phytohormone was almost non-detectable in the inactive AZs (Figure 7B), whereas strong green fluorescence, indicating its accumulation in the stressed floral AZ, was detected (Figure 7D,F). The signal was present inside the vascular bundles (Figure 7E). ABA was observed in drought-activated AZs, especially in the cells that had lost their adhesion (Figure 7F).

A similar tendency was observed for the expression of *LlACS* and *LlACO*, and the level of ACC. The values of the examined parameters increased in AZ during lupine development (Figure 8). Drought caused strong, 16-times higher transcriptional activity of *LlACS* in the flower AZ harvested from 48-day-old plants (Figure 8A). The expression was maintained at a similarly high level in the subsequent stage, as well. The ACC content in the AZ of the stressed plants was almost two-times higher than in the control (Figure 8B). ET precursor was strongly accumulated on day 51. An over five-time increase in the *LlACO* mRNA content as compared to the control plants was observed in the floral AZs of the 48-day-old plants cultivated under soil drought conditions (Figure 8C). Slightly larger transcript accumulation was found in the subsequent time-variant, while in the control AZs of 48-day-old plants the *LlACO* mRNA content was almost 50% lower than in the 51-day-old ones. Interestingly, the mRNA content of *LlACO* was lower than that of *LlACO* in all of the studied variants (Figure 9A,C). Soil drought stress caused ACC accumulation in the AZ cells (Figure 9E), vascular bundles (Figure 9D–F), and the neighboring cells (Figure 9F). On the cellular level, the signal was observed within the cytoplasm. Traces of fluorescence in the AZ cells, indicating the presence of ACC, were also found in the inactive AZs (Figure 9B).

## 3. Discussion

Drought leads to marked changes in the systemic level. The reduction in the number of leaves and their area limit transpiration and is one of the protective mechanisms adopted by plants. This effect has not only been observed in *L. luteus* (Appendix A), but also in *Abies alba*, *Picea abies*, *Fagus sylvatica,* and oak [12]. Soil drought leading to water deficit in the cells causes the stomata closure, which—together with disturbed photosynthesis—generates oxidative stress. H_2_O_2_ accumulation triggered by this stressor has been shown to occur in *Nicotiana tabacum*, *Citrus*, *Oryza sativa*, *Pisum sativum*, and *Gossypium hirsutum* [13,14]. Oxidative burst leads to damage on different organizational levels, e.g., chlorophyll degradation and disruption of the photosynthetic apparatus [15]. In yellow lupine, drought stress causes a reduction in the value of the Fv/Fm parameter (Appendix A) that describes the maximum quantum yield of photosystem II. A similar tendency has been observed during drought-stimulated leaf abscission in *Manihot esculenta* [16]. Photo-oxidation in *Portulaca oleracea*, *Phaseolus vulgaris*, and *Carthamus tinctorius* leads to a reduction in chlorophyll concentration and photosynthetic activity [17,18]. Analyses of *Pinus banksiana* seedlings have shown that the value of these parameters can be increased by applying ABA [19]. Yellow lupine cultivation under soil drought conditions leads to reduced leaf moisture (Appendix A) and significantly changes leaf contents of Fe, K, P, Zn, Na, and S (Appendix A). A decreasing level of iron, which is a component of cytochromes and ferredoxin, may lead to the disturbance of electron transport during photophosphorylation and in the respiratory chain. The deficiency of iron, an element of CAT and peroxidase, weakens the activity of antioxidant enzymes, limits plant growth, and, consequently, reduces yielding [20,21,22,23,24]. Iron deficiency may evoke a similar effect in lupine, which belongs to the same group as *Glycine max* for which high iron concentrations have been proven to increase yielding [25,26,27]. The disturbed water management observed in yellow lupine may be associated with its decreased amounts of potassium (Appendix A), which regulates stomata closure and affects plasmalemma and tonoplast permeability. Reduction in potassium content contributes to lower yielding [28], while the reduction in phosphorus concentration (Appendix A) may have a negative impact on photosynthesis, respiration, the function of cytoplasmic membranes, and resistance to stress. This is confirmed by research on *Phoebe zhennan*, which has shown phosphorus to stimulate a protective response to drought [29]. The zinc content variations observed in the leaves of older lupines (Appendix A) may be associated with their reduced leaf area (Appendix A). As for the reduction in sodium content in the leaves (Appendix A), it may worsen cellular colloid hydration, and thus have a negative impact on water management. Similarly to zinc, sulfur content variations caused by drought stress have only been observed in the leaves of 51-day-old plants (Appendix A). The results of studies performed by Abuelsoud et al. [30] have proven sulfur’s role in the protective response and have pointed to increased sulfur demand during drought stress.

The *Fabaceae* family, among them yellow lupine, are particularly sensitive to water deficits during flowering and pod setting. Yellow lupine flower separation occurs in the AZ located at the base of their pedicels [31]. The formation of this structure does not guarantee that the organ will separate, as it needs to be activated first, which is most often triggered by a combination of both endogenous and biotic or abiotic exogenous factors [4,32,33,34,35]. We showed that soil drought stress decreases AZ water content (Figure 1A), which may act as a signal for AZ activation. The results of our previous analyses indicated that the cells of a naturally active yellow lupine floral AZ underwent intensive divisions and were characterized by formation of new daughter cell walls, numerous plasmodesmata, vesicles, tiny aggregates, and large nuclei; whereas the observed changes pointed to the occurrence of the synthesis and transport of molecules that are responsible for ensuring cellular communication within the AZ and involved in early organ separation stages [11]. In this paper, we show that on the cellular level, soil drought stress causes changes (Figure 2E–H) that are similar to those found in naturally active AZs [11]. Resulting from middle lamella dissolution, tissue integrity is lost and continuity interrupted (Figure 2E,F), while the cells contain numerous grains, indicating high metabolic activity and AZ activation (Figure 2H), which is also reflected in an enhanced flower abortion rate after drought (Figure 1B). Following a reaction with Coomassie, an increased level of proteins was observed (Figure 2G,H) which, as other authors have suggested, must be synthesized de novo if organ abscission is to take place [35,36]. 

The anatomical and physiological symptoms occurring in the AZ are genetically controlled. It has been shown for *A. thaliana*, *Litchi chinensis,* and *Citrus* that AZ activation is controlled by the *IDA* and/or *IDA*-like genes [9,37,38,39,40,41]. *IDA* overexpression stimulates AZ cellular divisions, the excretion of large amounts of arabinogalactan, and premature organ separation, whereas *ida* mutants do not abscise their organs despite possessing well-developed AZs [37,42]. In *L. luteus*, progressive degradation processes in naturally active AZs are accompanied by *LlIDL* mRNA accumulation [11]. In our current study, we have shown that drought causes *LlIDL* transcript accumulation (Figure 3A). Additionally, this stress factor leads to an increase in the mRNA content of the newly-characterized *LlHSL* gene (Figure 3B). In *A. thaliana*, proline has been identified within the EPIP domain as undergoing post-translational modification to hydroxyproline and forming a covalent bond with glutathione at position 266 of the HAE/HSL2 receptor [43]. The ligand IDA binds with the extracellular part of the HAE/HSL2 receptor and triggers the pathway leading to AZ activation [10,44]. The predicted amino acid sequence of LlHSL (Appendix A) indicates that *LlHSL* cDNA encodes a receptor serine/threonine kinase located within the cell membrane. Within the LlHSL LEUCINE-RICH REPEAT RECEPTOR-LIKE KINASE PROTEIN (LRR-RLK) is found that is responsible for protein-protein interactions. Moreover, the central part of the domain contains an aspartic acid residue which is of significance to the enzyme’s catalytic activity (Appendix A). The *LlIDL* and *LlHSL* transcriptional activity patterns in yellow lupine floral AZs suggest that flower abscission may be activated by LlIDL within the LlHSL-triggering pathway. It confirms that the IDA–HAE/HSL mechanism is highly conserved in plants [45,46,47]. Research on other plant species has shown that the *HAE*/*HSL* mRNA content may be stimulated by ABA [48,49]. The positive feedback loop between HAE/HSL and AGAMOUS-like 15 (AGL15) has also been reported [9,44,49]. By inactivating the transcription of the AGL15 abscission repressor, the IDA–HAE/HSL complex-triggered MAPK cascade in the AZ and stimulate *HAE*/*HSL* expression [49,50]. In *L. luteus*, the elevated *LlIDL* and *LlHSL* transcriptional activity are accompanied by a high *LlMPK6* mRNA content (Figure 3C) and MPK6 accumulation in the pedicel vascular bundles and the AZ cells (Figure 4C,D,E). This suggests that MPK6 participates in AZ activation. In *A. thaliana*, the flowers of the *ida* or *hae* and *hsl2* mutants demonstrate decreased MPK6 activity, suggesting that it acts downstream [9]. Flower abscission correlated with increased expression of *LeMPK2,* a homolog of *AtMPK6*, has been observed in *Solanum lycopersicum* [51]. In summary, the *LlIDL*, *LlHSL,* and *LlMPK6* transcriptional activity patterns in the AZ show that the identified genes may encode succeeding elements of the pathway, regulating the time of flower separation. These processes are accompanied by oxidative stress conditions. Reactive oxygen species (ROS) have been shown multiple times to play an important role in cell wall modification processes occurring within the AZ. ROS cause hydrolysis of the polysaccharides of cell walls, and by generating local redox potential they affect the permeability and remodeling of plasmodesmata [52]. We previously indicated that AZ cells accumulate ROS following flower abscission in yellow lupine [6]. The appearance of ROS calls for a well-functioning antioxidative system including, inter alia, CAT responsible for H_2_O_2_ dismutation. In yellow lupine, drought changes H_2_O_2_ concentration (Figure 5H), increases CAT activity (Figure 5G), and affects the localization of this enzyme within selected areas of the AZ (Figure 5C–F)—which confirms the existence of stressful conditions and indicates modifications in the redox balance. Water deficit triggers a pathway in which MKK1-MPK6 modulate ABA-dependent *CAT1* expression in *A. thaliana* [53,54]. Furthermore, analyses of transcriptomes in *Manihot esculenta* have shown differential expression of the genes participating in ROS-induced pathways encoding superoxide dismutases and CAT [16].

One of the earliest metabolic responses of plants to the changing osmotic potential of the cells is endogenous ABA content increase, which is usually accompanied by higher resistance to stress [55,56,57,58,59,60]. This correlation has been observed in *Zea mays*, *Capsicum annum*, *Pinus banksiana,* and *Tradescantia virginiana* [19,61,62,63,64,65,66,67,68,69]. On the other hand, in *Phaseolus vulgaris,* drought stress does not increase the ABA content, but affects its transport from the symplast to the apoplast [70]. The ABA accumulation may be the result of elevated transcriptional activity of the gene encoding enzymes involved in its biosynthesis, e.g., *ZEP*. It has been shown many times that the expression of this gene is stimulated by stress factors, including drought [71,72]. In yellow lupine, soil water deficit increases the *LlZEP* mRNA content in the AZ (Figure 6A). Our analyses are the first to determine the changes of the gene’s transcriptional activity in this small fragment of tissue under drought stress, suggesting local control of ABA biosynthesis and confirming the specificity of small AZ fragments. Most of the former analyses concentrated on determining the expression of *ZEP* in particular organs affected by stress, e.g., the root, the shoot, or the leaves. In *N. plumbaginifolia* and *Solanum lycopersicum*, drought has been shown to increase *ZEP* transcriptional activity in the roots, while ABA to accumulate not only in the roots, but also the leaves [71,73]. In *Vigna unguiculata*, the same stressor has failed to affect the mRNA content of *VuABA1*, a gene that encodes zeaxanthin epoxidase; in *A. thaliana* it causes *ZEP* accumulation in the stems and roots, and in *Medicago sativa* the *MsZEP* transcript content is lower in the shoots, while its expression patterns in the roots are diverse [56,74,75]. The results of transcriptomic analyses in *Litchi chinensis* have shown a correlation between fruit abscission and elevated expression of *ZEP* and the ABA receptor-encoding gene—*PYR*/*PYL* (*pyrabactin resistance*/*PYR1-like*) [76]. In yellow lupine, the changes in the transcriptional activity of *LlZEP* in the AZ of flowers cultivated under water deficiency conditions are accompanied by increasing content of ABA (Figure 6B), and by its localization in the AZ cells (Figure 7D–F). In our previous studies, we observed a similar tendency in naturally active AZs [11]. The mechanism of ABA activity is based on the triggering of the pathway of kinases involved in cell signaling (e.g., SnRK2, SAPK2) and regulation of the synthesis of low-molecular-weight hydrophilic and thermolabile protective proteins, such as aquaporins (PIP1, PIP2), osmotins, or dehydrins, which optimize water binding and improve the ion-selective properties of the cell membrane [77,78,79,80,81]. Moreover, ABA stimulates the activity of enzymes responsible for the synthesis of osmoprotectants, such as proline, betaine, and glycine, which protect cell structures from the effects of dehydration [82].

In our previous research, we have shown that flowering abortion in yellow lupine evoked by ABA resulted from its stimulatory effect on the expression of the two main ET biosynthesis genes, *LlACS* and *LlACO*, as well as ACC content [5]. Presented qPCR analyses have shown that soil drought stimulates the transcriptional activity of *LlACS* in the AZs of yellow lupine flowers (Figure 8A) and elevates the content of ACC (Figure 8B), which is localized in the AZ (Figure 9). The accompanying high expression of *LlACO* (Figure 8C) may indicate that ACC is oxidized to ET, which is the main abscission effector responsible for activating cell wall hydrolytic enzymes [83,84]. The role of soil drought in regulating the expression of *ACS* and *ACO,* and the production of ACC and ET, has been proven in research on *Citrus reshni* [85]. In the root, drought led to the accumulation of ACC, which was then transported by xylem to the above ground plant parts and oxidized in a reaction catalyzed by ACO to ET. This effect was not only associated with the increased levels of ET in the tissues, but also a heightened sensitivity of the cells to that phytohormone caused by the changes in the concentrations of auxins or ABA [33]. Leaf abscission in citrus evoked by increasing ACC levels was accompanied by elevated ET production [86]. An important role in regulating the contents of precursors in the individual organs is played by their translocation. In the cell, ACC can be transported across the tonoplast by ATP-dependent transporters [87]. On the systemic level, under soil drought conditions, ET precursor is transported by elements of the vascular bundles, with this mechanism having been thoroughly described for *Solanum lycopersicum* and *Citrus reshni* [85,88]. It has been found, as well, that in *Gossypium hirsutum,* ACC can be transported by the phloem [89]. It appears that ACC may be one element of long-distance pathways triggered in response to drought stress [90]. Thus, there exist two possible interpretations of the results obtained for yellow lupine. ACC either plays the role of a signaling molecule in generating the plant’s response to water deficit, or—in an ACO-catalyzed reaction—it enables the formation of ET which, by regulating the expression of genes encoding hydrolytic enzymes, leads to AZ activation.

Our study provides information about several new potential markers of early response to water loss, which can help elucidate the mechanisms controlling plant response to drought, and forms a useful basis for breeders and agronomists to enhance crop tolerance to stress. Based on our results, we have proposed a floral AZ activation model for *L. luteus* (Figure 10).

## 4. Materials and Methods

### 4.1. Plant Material and Growth Conditions

*Lupinus luteus* (L.) Taper variety (2n = 52) was used in the experiments. All plants were grown in phytotron chambers in controlled conditions, which have been optimized and described previously by Frankowski et al. [31]. The lupines (200 plants) were watered for five weeks in optimal 70% soil water holding capacity (WHC); subsequently, part of them (100 plants) were grown for two weeks in the 25% WHC. For calculating WHC, the modified method of Chauhan and Johnson (2011) was used [91]. The pots (11 L) containing soil material were saturated with tap water. The pot surface was then covered with a plastic container and the pots were allowed to drain for 48 h. Thereafter, from the middle of each pot, three soil samples (each ~500 g) were taken. These samples were weighed (wet weight of soil, A), oven-dried (90 °C for 72 h), and reweighed (dry weight of soil, B). The WHC was then calculated using the formula: [(A-B) × 100]/B. 

### 4.2. Sample Harvest

The plant material for analysis was selected after 48 and 51 days of plant growth in soil with optimal moisture (70% WHC) and from drought conditions (25% WHC). The flower pedicels and stem fragments (~2 mm thickness only) containing abscission zone (AZ) (Appendix A) were used as the experimental material. AZ sections of stressed and unstressed plants were dissected using a razor blade under a binocular microscope (Appendix A). We applied the methodology previously described by Frankowski et al. [7]. For the purpose of expression and gas chromatography mass spectrometry (GC-MS), analyses tissues (0.2 g or 0.5 g, respectively) were frozen in liquid nitrogen and stored at −80 °C until use, while for histological and immunocytochemical studies, 15 AZs from each variant were immediately fixed. In turn, 2 g of leaves (Appendix A) for Inductively Coupled Plasma—Optical Emission Spectrometers (ICP-OES) studies were sampled and dried at 60 °C for 24 h. 

### 4.3. Evaluation of Stress Impact on Plants

Relative water content in AZ and leaves harvested after 48 and 51 days of plant growth in drought (25% WHC) and soil with optimal moisture (70% WHC, control plants) was measured using the drying method. Plant material (2 g) was weighed (analytical weight PB221S, Sartorius, an accuracy of ± 0.0001 g), placed in zinc vessels, and dried (SUP200W, WAMED) (60 °C) for 24 h. Then, plant material was cooled down in desiccator and weighed again. The relative water content percent (RWC) was determined through the following equation: RWC [%] = (FW-DW)/SW × 100, where FW: Fresh gross weight [g]; DW: Dry gross weight [g]; and SW: Sample weight [g]. The dried material was also used for element content determination. 

The influence of drought stress on flower abortion rate was presented as a percentage of separated flowers in relation to the flowers formed by the plant. Additionally, we determined the effect of drought on the number of leaves per plant and leaf area per plant using AM350 Portable Leaf Area Meter (ADC BioScientific Ltd., Hoddesdon, UK). The maximum quantum efficiency of PS II (Fv/Fm) as a reliable marker of photo-inhibition [92] was measured using an OS-30P (Opti-Sciences, Inc., Hudson, NH, USA) according to the methods described by Weng (2006) [93]. The results were presented as the mean of the sum of the areas of all leaves per plant. 

### 4.4. Elements Determination

Dried leaves (0.1–0.5 g) were homogenized in a mortar with a pestle and mixed with 8 mL of HNO_3_ (65%, *v*/*v*) and 1 mL of HCl (37%, *v*/*v*). Next, the samples were microwaved (Multiwave 3000, Anton Paar) under the following conditions: Parameter stabilization (p-Rate-0.5 bar s^−1^, 190 °C, 40 bar, 1050 W) for 10 min, mineralization for 20 min, cooling up to 50 °C for 20 min. The samples were diluted with water to 100 mL. For Fe, K, P, Na, S, and Zn determination, the ICP-OES (Optima 7300 DV, Perkin Elmer, Waltham, MA, USA) was used. The linearity of calibration curves was established for Fe, K, P, Na, S, and Zn (Appendix A). The wavelength of the emission of individual element measurement is shown in Appendix A. All examined samples contained reference sample Hs-12. Argon was served as a carrier gas. The results were analyzed by WinLab 32 (Perkin Elmer) software.

### 4.5. Preparation of AZ for Microscopic and Histological Studies 

The dissected AZ fragments were fixed in 4% paraformaldehyde (*w*/*v*) and 0.25% glutaraldehyde (*v*/*v*) prepared in 1× PBS (pH 7.2) at 4 °C overnight. Tissue fragments were then washed 3 times in 1× PBS (pH 7.2), dehydrated in an ethanol series, supersaturated, and embedded in BMM resin (butyl methacrylate, methyl methacrylate, 0.5% (*w*/*v*) benzoin ethyl ether, 10 mM dithiothreitol; Fluka, Buchs, Switzerland) at −20 °C using UV light for polymerization, according to Wilmowicz et al. [5]. Semi-thin sections (1.5 μm) were obtained with an Ultracut microtome (Reichert Jung, Vienna, Austria). They were placed on slides covered with BioBond (BB International, Cardiff, UK) and used for histochemical staining and immunolocalization studies.

For histological features and determination of proteins, AZs were stained with 0.05% Toluidine blue O (1.15930, Sigma-Aldrich, St. Louis, MO, USA) and Coomassie Brilliant Blue G 250 (27816, Sigma-Aldrich), respectively. Next, sections were observed in the LM Zeiss Axioplan (Carl Zeiss, Oberkochen, Germany) microscope equipped with a ProGres C3 digital camera. Micrographs were obtained by using the ProGres CapturePro 2.6 software (Jenoptik AG, Jena, Germany).

### 4.6. Immunolocalization of ABA, ACC, MPK6, and CAT

The AZ sections for immunofluorescence studies were prepared according to Wilmowicz et al. [5]. The sections on slides were blocked in 1% bovine serum albumin (BSA) for 2 h and incubated with primary antibody anti-ACC (AS11 1800, Agrisera) and anti-ABA (AS06195, Agrisera), and subsequent steps were performed according to the protocol described by Wilmowicz et al. [5]. For immunolocalization of MPK6 and CAT, the sections were incubated overnight at 4 °C in primary antibody solution prepared by dilution the primary antibody anti-MPK6 (AS12 2633, Agrisera) and anti-CAT (AS09 501, Agrisera) 1:25 in 1% BSA dissolved in 1× PBS (pH 7.2). A DyLight Alexa 488 conjugated IgG diluted 1:250 in PBS buffer for 2 h at 37 °C was served as the secondary antibody (AS09 633, Agrisera). Negative control reaction required for validation of the immunohistochemical findings was performed by omitting the incubation with the primary antibody, and showed no labeling (Appendix A).

### 4.7. RNA Isolation and cDNA Synthesis

Frozen material (100 mg fresh weight) was homogenized in a sterile chilled mortar with a pestle. Total RNA was extracted with the NucleoSpin RNA Plant, MACHEREY-NAGEL GmbH & Co. KG (Düren, Germany) according to the manufacturer’s instructions. First strand cDNA was synthesized from 1 μg of RNA primed with anchored oligo(dT)_18_ primers with the Transcriptor First Strand cDNA Synthesis Kit (ROCHE Diagnostics GmbH, Mannheim, Germany) following manufacturer’s guidelines.

### 4.8. Isolation of LlZEP, LlHSL, and LlMPK6 cDNAs

The full length cDNA of *LlZEP* was obtained by the PCR reactions with degenerated primers (Appendix A), performed in the T3 Thermocycler (Biometra, Göttingen, Germany). Reaction’s conditions, described previously by Frankowski et al. (2015b), were applied. Amplified cDNA fragments (Appendix A) were isolated from an agarose gel with the GeneMATRIX Agarose Out DNA Purification Kit (EurX, Gdańsk, Poland) and cloned using the Strata Clone PCR Cloning Kit (Agilent Technologies, Santa Clara, CA, USA), according to manufacturer’s recommendations. Plasmid DNA was purified with the GeneMATRIX PLASMID MINIPREP DNA Purification Kit (EurX, Gdańsk, Poland) and sequenced by “Genomed S.A.” (Warsaw, Poland). A full-length cDNA for *LlZEP* was obtained using the BD SMART RACE cDNA Amplification Kit (Clontech, Mountain View, CA, USA) and 5′-RACE and 3′-RACE primers (Appendix A). The RACE-PCR reactions were performed using the Advantage 2 PCR Enzyme System (Clontech-Takara Bio Europe, Saint-Germain-en-Laye, France). The obtained products were purified from agarose gel (Appendix A) and cloned using the Strata Clone PCR Cloning Kit. Sequence data (Appendix A) has been deposited at the GenBank database.

The full-length *LlHSL* and partial *LlMPK6* cDNAs sequence (Appendix A) were obtained by the traditional PCR methods with specific primers listed in Appendix A, constructed for fragments of *HSL* and *MPK6* from *L. luteus* deposited in SRA database under the accession number PRJNA285604 (BioProject), with the experiment accession number SRX1069734.

### 4.9. RT-qPCR

The expression of *LlACS* (GenBank acc. no KF573522), *LlACO* (GenBank acc. no KF573523), *LlZEP, LlIDL* (GenBank acc. no KT716180)*, LlHSL,* and *LlMPK6* genes was analyzed in relation to *LlACT* (GenBank acc. no KP257588), which was used as a reference endogenous control for normalization purposes. RT-qPCR was performed with a LightCycler 2.0 Carousel-Based System (ROCHE Diagnostics GmbH, Germany) and a LightCyclerTaqMan Master Kit (ROCHE Diagnostics GmbH, Germany), using primers and UPL probes designed previously [5,7]. The list of gene-specific and reference primers used is presented in Appendix A. The qPCR mixture preparation and conditions were as described previously [7]. Reaction efficiencies (>99%) were calculated based on the standard curves from serial dilutions of cDNA templates, and relative expression values were obtained by LightCycler Software 4.1 (Roche Diagnostics GmbH, Mannheim, Germany). The data are presented as means ± standard deviation (SD) of three biological repeats obtained from three independent experiments.

### 4.10. ABA and ACC

For the determination of endogenous ABA and ACC, a GC-MS was used. The AZ fragments (0.5 g) were powdered and subsequently, ABA and ACC content were analyzed according to protocols optimized by Wilmowicz et al. [5] and Kućko et al. [6], respectively. GC-MS-SIM was performed by monitoring m/z 245 for phthalimido-ACC-methyl ester and m/z 249 for deuterated phthalimido-ACC-methyl ester and m/z 162 and 190 for ABA and 166 and 194 for [6-^2^H_3_]ABA.

### 4.11. Hydrogen Peroxide Detection and CAT Activity Measurements

The concentration of H_2_O_2_ was analyzed using Shimadzu spectrophotometer (UV-160 1PC). Firstly, 500 mg tissue was homogenized with 5 mL 1% trichloroacetic acid (TCA) solution. Homogenates were centrifuged (16,000× *g*) for 10 min at 4 °C. Obtained supernatants were transferred to new Falcon tubes and brought to pH 7.5 (4 N KOH). Next, the samples were centrifuged (10,000× *g*) for 1 min, and 1 mL of supernatant was transferred to new tubes and mixed with 250 µL DMAB (19.8 mM) in 0.5 M phosphate buffer (pH 6.5), 230 µL MBTH (0.456 mM), and 20 µL peroxidase (0.25 U). The mixtures were vigorously shaken for 20 min at 25 °C. Then absorbance was measured (590 nm) and the total H_2_O_2_ content was calculated from the calibration curve. The results were presented as µmol H_2_O_2_ per gram of fresh weight.

CAT activity of the soluble protein extracts was analyzed by the method described by Beers and Sizer with some modifications [94]. For protein isolation, AZs (~0.5 g) were homogenized in liquid N_2_, extracted in 1 mL of 0.05 M potassium phosphate buffer, and mixed for 1 h at 4 °C. Then vials were centrifuged (13,500× *g*) for 1 h at 4 °C. The supernatant was used for subsequent analyses. Homogenate was transferred into an eppendorf tube and filled with extraction buffer to the final volume of 2 mL. Samples were centrifuged (20 min, 20,000× *g*, 4 °C). The supernatant was used for enzyme assay. Protein content was quantified according to the Bradford method [95]. Next, 100 μL of the extract was added to 1.0 mL of 20 mM H_2_O_2_ in 50 mM potassium phosphate buffer (pH 7.0), and H_2_O_2_ decomposition was monitored at 240 nm [94]. One unit of CAT activity catalyzed the degradation of 1 µmol of H_2_O_2_ per min. The results were presented as the average ± SD CAT activity from three independent replications (*n* = 3).

### 4.12. Statistical Analysis

All data are the results of three biological samples (one sample was the mix of AZ fragments from different plants growing at the same time) with two technical replications (each biological sample was analyzed two times) (*n* = 6) and presented as mean ± standard error (SE). Statistical data were obtained by Microsoft Excel, whereas the statistics program SigmaPlot 2001 v.7.0 was used for statistics and generation for the graphs.

## Figures and Tables

**Figure 1 ijms-20-03731-f001:**
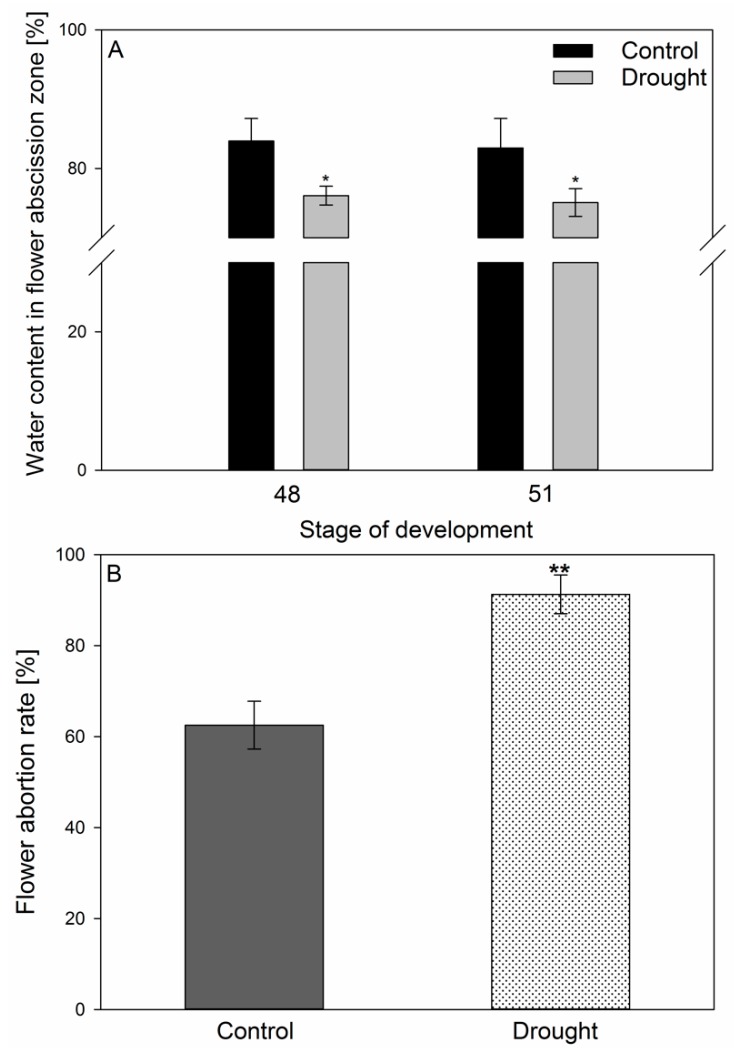
The influence of soil drought stress on the water content in flower abscission zone (AZ) [%] (**A**) and flower abortion rate (%) (**B**) in *Lupinus luteus*. Control plants were growing in soil of optimal moisture (70% holding capacity; WHC). In parallel, part of plants was subjected to drought conditions for 2 weeks (25% WHC). Plant material for water content measurements was collected on the 48th and 51st days of cultivation (**A**). The number of aborted flowers per one lupine was counted and the data were presented as averages of 15 technical replicates ± SE. Significant differences to stress plants in comparison to control are indicated as * *p* < 0,05, ** *p* < 0.01 (Student’s *t*-test).

**Figure 2 ijms-20-03731-f002:**
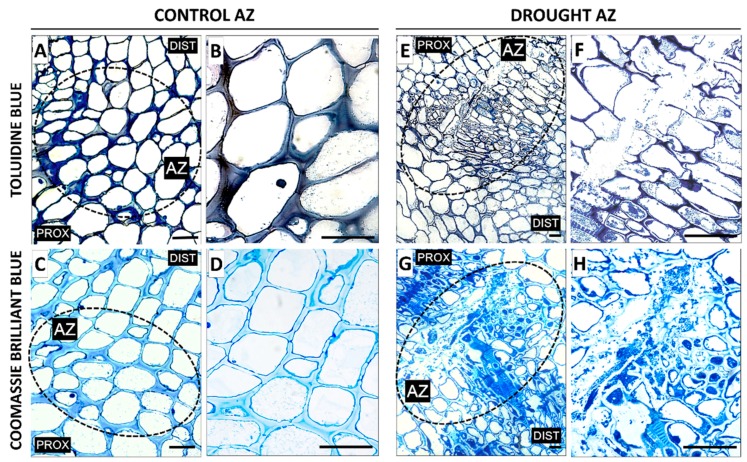
The impact of soil drought stress (25% WHC) on the structural features in the flower abscission zone (AZ) of *Lupinus luteus*. For observations, sections of AZ were excised on the 48th day of cultivation. Samples of AZ from control and drought-stressed plants were stained with toluidine blue (**A**,**B**,**E**,**F**) or Coomassie Brilliant Blue (**C**,**D**,**G**,**H**). Images B, D, F, and H are magnifications of different areas presented in A, C, E, G, respectively. AZ area was marked by black dotted circle (**A**,**C**,**E**,**G**). Abbreviations: PROX—stem fragments below the AZ, DIST—flower pedicel fragments above the AZ. Scale bars: 40 µm.

**Figure 3 ijms-20-03731-f003:**
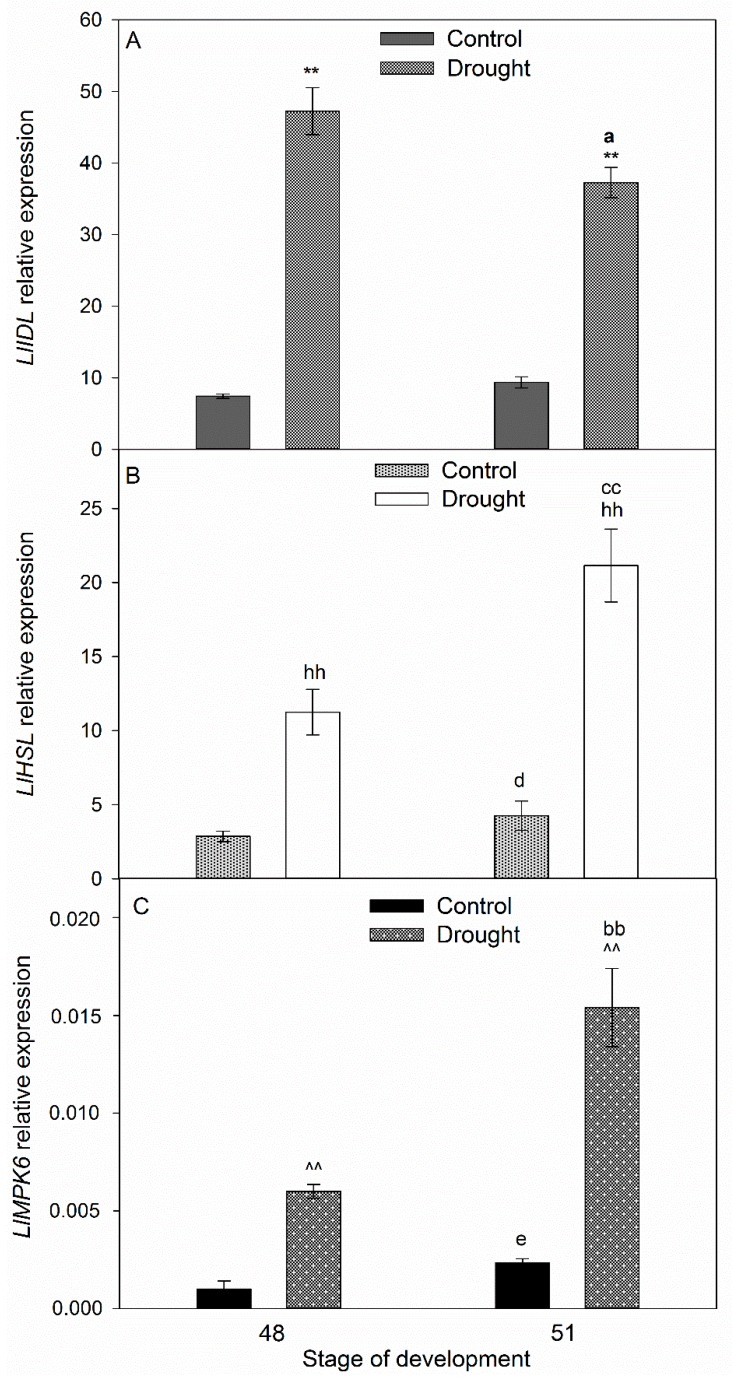
Transcriptional activity of *LlIDL* (**A**), *LlHSL* (**B**), and *LlMPK6* (**C**) (related to *LlACT*) in the yellow lupine flower abscission zone (AZ) subjected to drought. Control plants were growing in soil of optimal moisture (70% WHC), while part of plants was subjected to drought conditions for 2 weeks (25% WHC). For gene expression profiling, AZs were harvested on the 48th and 51st days of cultivation. Data are presented as averages ± SE. For *LlIDL* expression, significant differences in stressed plants in comparison to control plants are indicated as ** *p* < 0.01, and significant differences in 51-day-old stressed plants in comparison to 48-day-old plants are indicated as ^a^
*p* < 0.05. For *LlHSL,* significant differences in stressed plants in comparison to control plants are indicated as ^hh^
*p* < 0.01, significant differences in 51-day-old control plants in comparison to 48-day-old plants are indicated as ^d^
*p* < 0.05, and significant differences in 51-day-old stressed plants in comparison to 48-day-old stressed plant are indicated as ^cc^
*p* < 0.01. For *LlMPK6,* significant differences in stressed plants in comparison to control plants are indicated as ^^^^
*p* < 0.01, significant differences in 51-day-old control plants in comparison to 48-day-old plants are indicated as ^e^
*p* < 0.05, and significant differences in 51-day-old stressed plants in comparison to 48-day-old stressed plants are indicated as ^bb^
*p* < 0.01 (Student’s *t*-test).

**Figure 4 ijms-20-03731-f004:**
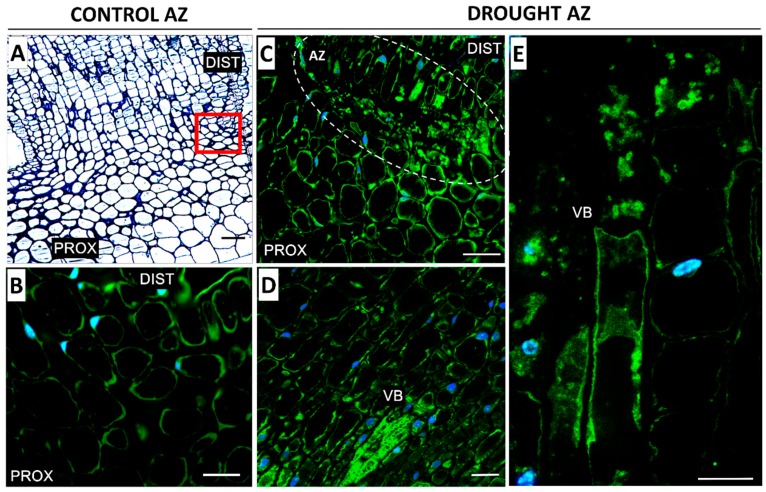
Immunofluorescence localization of MPK6 in the abscission zone (AZ) of yellow lupine flowers grown under drought conditions (**C**–**E**) and in the control AZ (**B**). The AZs were excised on the 48th day of development. Control plants were cultivated under optimal soil conditions (70% WHC), whereas stressed plants were subjected to drought for 2 weeks (25% WHC). The AZ region of stressed plants was highlighted by white lines (**C**). Images **D** and **E** present the vascular bundle area. Nuclei were stained with DAPI. The examined region of control AZ used in the immunofluorescence studies os indicated by a red square (**A**). Abbreviations: PROX—stem fragments below the AZ, DIST—flower pedicel fragments above the AZ, VB—vascular bundles. Scale bars: 100 µm (**A**, **C**), 40 µm (**B**,**D**,**E**).

**Figure 5 ijms-20-03731-f005:**
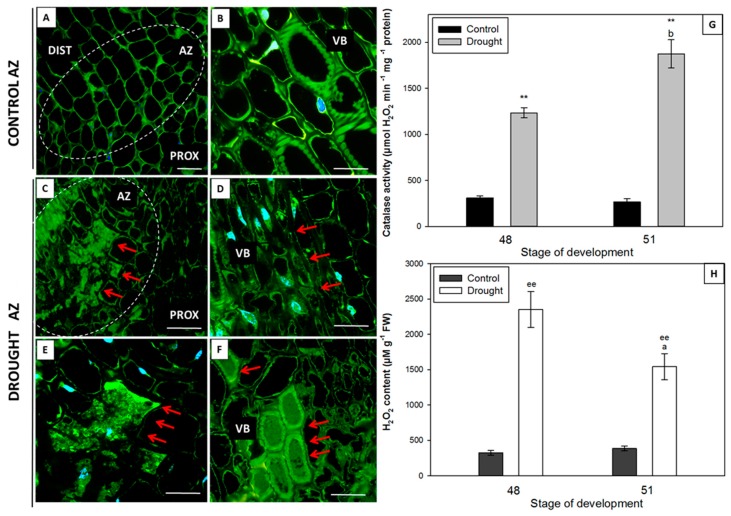
Immunolocalization of catalase (CAT) in the floral abscission zone (AZ) of yellow lupine grown in drought conditions (**C**–**F**) and in the AZ of control plants (**A**,**B**). The AZs were excised on the 48th day of development. Control plants were cultivated under optimal soil conditions (70% WHC). Stressed plants were subjected to drought for 2 weeks (25% WHC). The AZ region is indicated by white curves (**A**,**C**). The presence of CAT is highlighted by red arrows. Image **E** is magnified **C**. Nearby cells to vascular bundles (**D**). Magnified vascular bundles (**B**,**F**). Nuclei were stained with DAPI. Abbreviations: PROX—stem fragments below the AZ, DIST—flower pedicel fragments above the AZ, VB—vascular bundles. Scale bars: 100 µm (**A**,**C**), 60 µm (**D**,**F**), 40 µm (**B**,**E**). CAT activity (**G**) and hydrogen peroxide (H_2_O_2_) concentration (**H**) in the floral AZ of yellow lupine grown in drought conditions and in the AZ of control plants. Data are presented as averages ± SE. For CAT activity, significant differences in stressed plants in comparison to control plants are indicated as ** *p* < 0.01, and in 51-day-old stressed plants in comparison to 48-day-old plants are indicated as ^b^
*p* < 0.05. For H_2_O_2_ content, significant differences in stressed plants in comparison to control plants are indicated as ^ee^
*p* < 0.01, and in 51-day-old stressed plants in comparison to 48-day-old plants are indicated as ^a^
*p* < 0.05.

**Figure 6 ijms-20-03731-f006:**
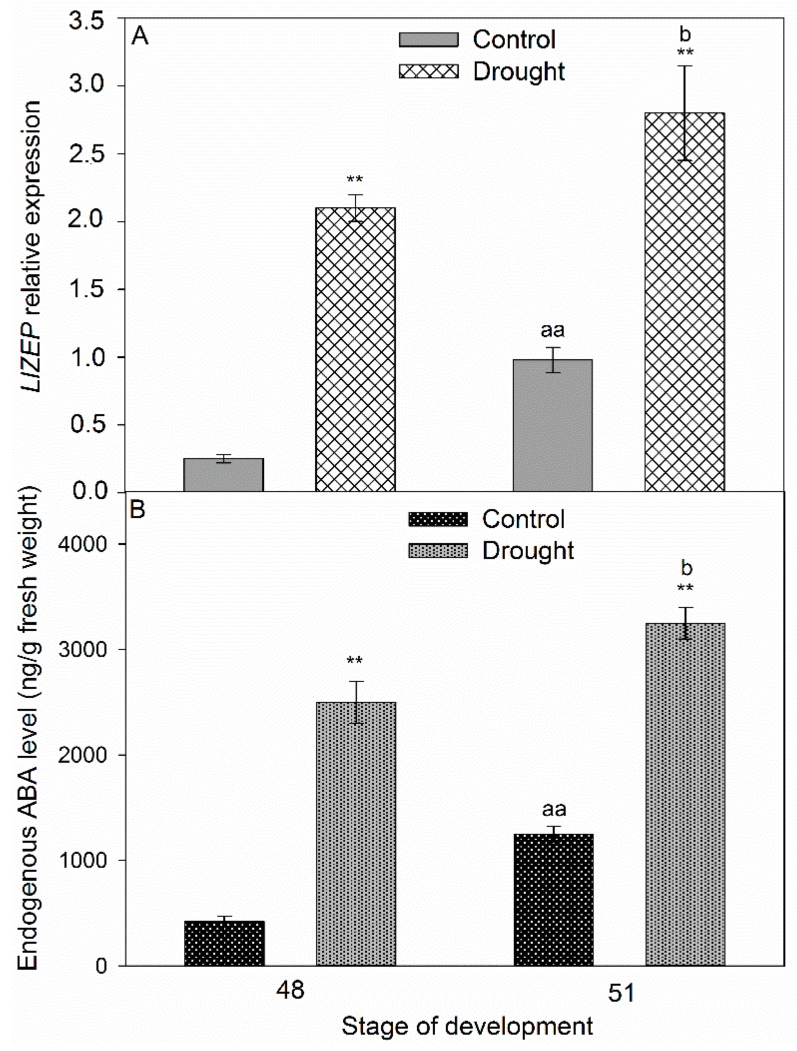
*LlZEP* expression (related to *LlACT*) (**A**) and endogenous content of abscisic acid (ABA) in floral abscission zone (AZ) of *Lupinus luteus* grown under drought conditions. Control plants were growing in soil of optimal moisture (70% WHC). In parallel, part of plants was subjected to drought for 2 weeks (25% WHC). For analysis, AZs were harvested on the 48th and 51st days of cultivation. Data are presented as averages ± SE. For *LlZEP* expression, significant differences in stressed plants in comparison to control plants are indicated as ** *p* < 0.01, significant differences in 51-day-old control plants in comparison to 48-day-old plants are indicated as ^aa^
*p* < 0.01, and significant differences in 51-day-old stressed plants in comparison to 48-day-old stressed plant are indicated as ^b^
*p* < 0.05. For ABA content, significant differences in the stressed plants in comparison to control plants are indicated as ** *p* < 0.01, significant differences in 51-day-old control plants in comparison to 48-day-old plants are indicated as ^aa^
*p* < 0.01, and significant differences in 51-day-old stressed plants in comparison to 48-day-old stressed plants are indicated as ^b^
*p* < 0.05 (Student’s *t*-test).

**Figure 7 ijms-20-03731-f007:**
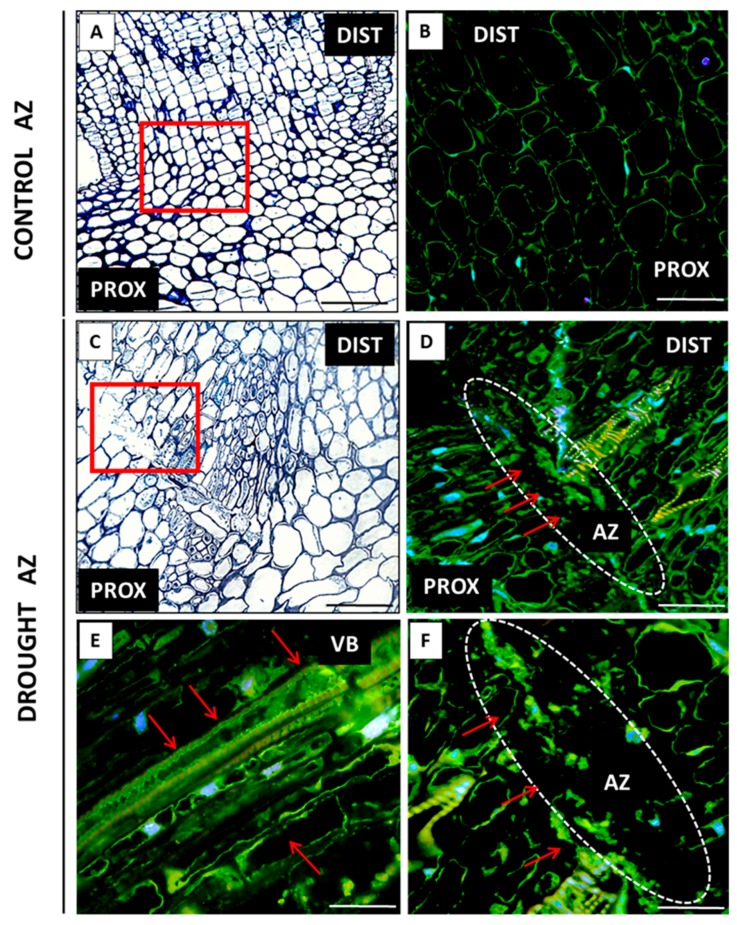
Tissue and subcellular localization of abscisic acid (ABA) in the abscission zone (AZ) of *Lupinus luteus* flowers grown in drought conditions (**D**–**F**) and in the AZ of control plants (**B**). The AZs were excised on the 48th day of development. Control plants were cultivated under optimal soil conditions (70% WHC). Part of plants was subjected to drought conditions for 2 weeks (25% WHC). The AZ regions are indicated by white curves (**D**,**F**). The presence of ABA (**D**–**F**) is highlighted by red arrows. Image **F** is magnified **D** region. Image **E** corresponds to the vascular bundles’ area. DAPI was used for nuclei staining. The examined regions of AZs used for the immunofluorescence studies are indicated by red squares (**A**,**C**). Abbreviations: PROX—stem fragments below the AZ, DIST—flower pedicel fragments above the AZ, VB—vascular bundles. Scale bars: 100 µm (**A**,**C**,**D**), 60 µm (**B**), 40 µm (**E**,**F**).

**Figure 8 ijms-20-03731-f008:**
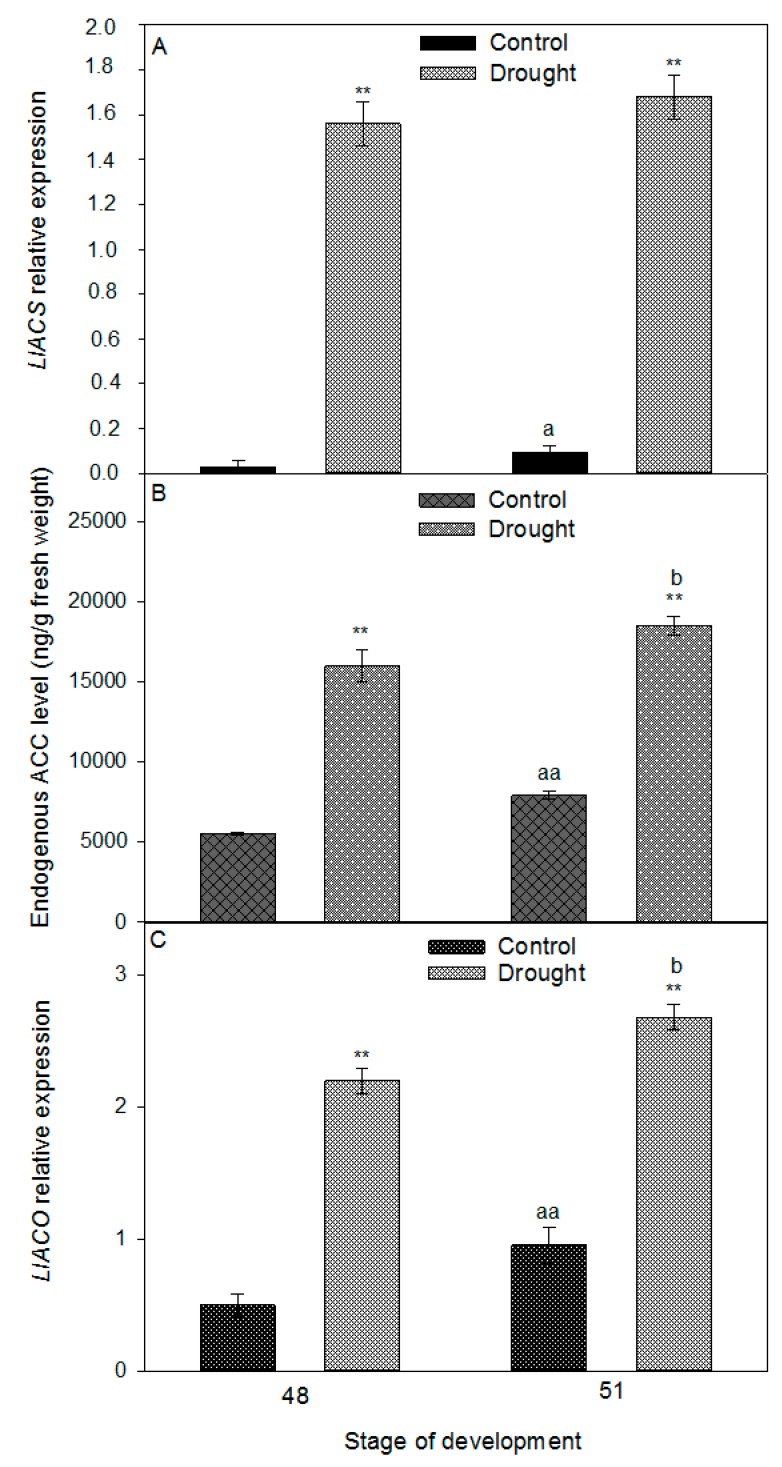
Expression analysis of ethylene (ET) biosynthesis genes, *LlACS* (**A**), and *LlACO* (**C**), (related to *LlACT*) and endogenous level of ET precursor-ACC in floral abscission zone (AZ) of *Lupinus luteus* grown under drought conditions. Control plants were cultivated in soil of optimal moisture (70% WHC). Part of plants was subjected to drought conditions for 2 weeks (25% WHC). AZs were harvested on the 48th and 51st days of cultivation. Data are presented as averages ± SE. For *LlACS* expression, significant differences in stressed plants in comparison to control plants are indicated as ** *p* < 0.01, and significant differences in 51-day-old control plants in comparison to 48-day-old plants are indicated as ^a^
*p* < 0.05. For *LlACO* expression, significant differences in stressed plants in comparison to control plants are indicated as ** *p* < 0.01, significant differences in 51-day-old control plants in comparison to 48-day-old plants are indicated as ^aa^
*p* < 0.01, and significant differences in 51-day-old stressed plants in comparison to 48-day-old stressed plant are indicated as ^b^
*p* < 0.05. For ACC content, significant differences in stressed plants in comparison to control plants are indicated as ** *p* < 0.01, significant differences in 51-day-old control plants in comparison to 48-day-old plants are indicated as ^aa^
*p* < 0.01, and significant differences in 51-day-old stressed plants in comparison to 48-day-old stressed plant are indicated as ^b^
*p* < 0.05 (Student’s *t*-test).

**Figure 9 ijms-20-03731-f009:**
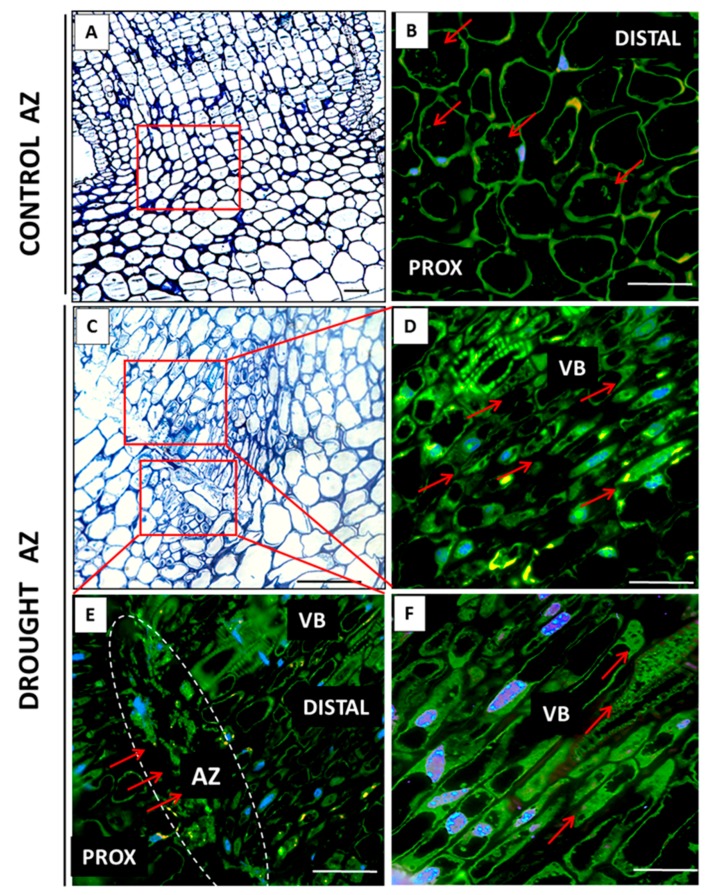
Immunolocalization of ethylene precursor, ACC, in the floral abscission zone (AZ) of yellow lupine grown in drought conditions (**D**, **E**, **F**) and in the AZ of control plants (**B**). The AZs were excised on the 48th day of development. Control plants were cultivated under optimal soil conditions (70% WHC), part of plants was subjected to drought conditions for 2 weeks (25% WHC). The AZ region is highlighted by white curves (**E**). The presence of ACC is indicated by red arrows. **D** and **F** correspond to vascular bundle areas. Nuclei were stained with DAPI. The studied regions of AZs used for the immunofluorescence studies are marked by red squares (**A**,**C**). Abbreviations: PROX—stem fragments below the AZ, DIST—flower pedicel fragments above the AZ, VB—vascular bundles. Scale bars: 100 µm (**A**,**C**), 60 µm (**D**,**E**), 40 µm (**B**,**F**).

**Figure 10 ijms-20-03731-f010:**
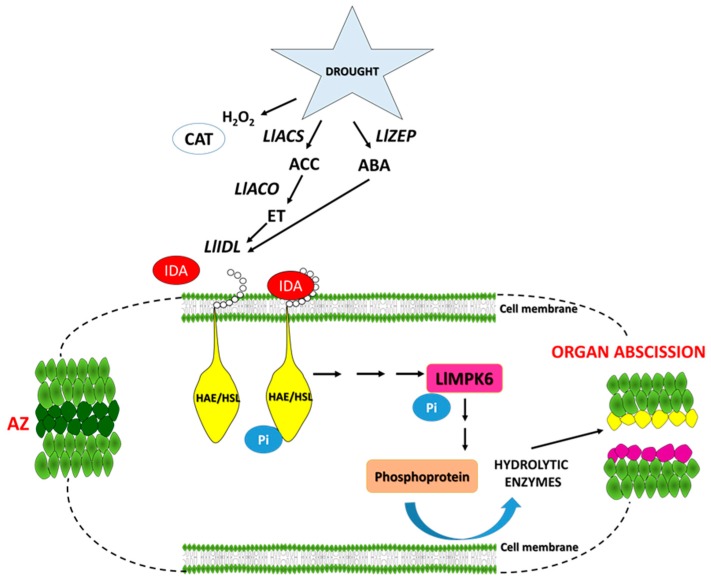
Hypothetical model of floral abscission zone functioning under drought stress in yellow lupine. The precise description is in the text.

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
