# Peer review of "Molecular and Hormonal Aspects of Drought-Triggered Flower Shedding in Yellow Lupine"

_ijms, 2019, doi:10.3390/ijms20153731_

Round 1
Reviewer 1 Report
This manuscript reported the molecular and hormonal changes in yellow lupine flower abscission zone (AZ) under drought stress. Drought condition lead to plant morphological changes, such as fewer leaves, smaller leaf area, less photosynthesis activity and decreased leaf water content. It also caused significant changes of elements composition, and higher flower abortion rate. Drought-stressed plant exhibited dissolution of middle lamellae in the AZ are, and higher protein contents in AZ cells. The introduction of drought stress induced transcriptional expression of IDA/HAE/MPK6 pathway, ZEP, ACS, ACO genes, and also increased endogenous ABA and ACC level.
The manuscript is intriguing and logically consistent. However, it missed two pieces in the flow. The authors showed the increased CAT activity in AZ cells by immunofluorescence, but did not show CAT gene expression level and endogenous ROS level (such as H2O2 content). On the other hand, it showed ACS and ACO gene expression levels, and the immunolocalization of ABA and ACC content, but did not show ZEP, ACS and ACO enzyme activity. It may be necessary to check these two things to make sure that it is a complete model.
Figure 3C, 6A, 8A, Supplemental Figure 8C, 10C, 11A and 11B, change comma in the Y-axis to decimal point.
Therefore, I suggest this manuscript go for a minor revision.
Author Response
Response to Reviewer 1 Comments
This manuscript reported the molecular and hormonal changes in yellow lupine flower abscission zone (AZ) under drought stress. Drought condition lead to plant morphological changes, such as fewer leaves, smaller leaf area, less photosynthesis activity and decreased leaf water content. It also caused significant changes of elements composition, and higher flower abortion rate. Drought-stressed plant exhibited dissolution of middle lamellae in the AZ are, and higher protein contents in AZ cells. The introduction of drought stress induced transcriptional expression of IDA/HAE/MPK6 pathway, ZEP, ACS, ACO genes, and also increased endogenous ABA and ACC level.
Point 1: The manuscript is intriguing and logically consistent. However, it missed two pieces in the flow. The authors showed the increased CAT activity in AZ cells by immunofluorescence, but did not show CAT gene expression level and endogenous ROS level (such as H2O2 content).
Response 1: The analysis of CAT and ROS state wasn’t the principal scientific problem to investigate in the Ms. We aimed to present CAT localization as an indicator of environmental stress only. That’s why we showed the localization of CAT in AZ. Nevertheless, we tried to improve our Ms according to Reviewer suggestion and we added the CAT activity under drought stress conditions (Fig. 5G). We also performed an analysis of H2O2 content following drought (Fig. 5H). Unfortunately, the coding sequence of CAT gene in yellow lupine has not been identified yet. For now, it’s not possible to conduct expression experiments. Bearing in mind the results obtained here, that revealed CAT involvement in the processes taking place in the AZ cells we will surely try to identify full length sequence of gene encoding CAT.
Point 2: On the other hand, it showed ACS and ACO gene expression levels, and the immunolocalization of ABA and ACC content, but did not show ZEP, ACS and ACO enzyme activity. It may be necessary to check these two things to make sure that it is a complete model.
Response 2: The proposed direction of scientific research is very interesting, however, quite challenging at the moment. Unfortunately, we haven’t optimized the method for activity determination in our lab and it is not possible to perform these experiments for now. We have found several assays for measurements of activity of these enzymes in other species (e.g. Dandekari AM, Teo G, Defilippi BG, Uratsu SL, Passey AJ, Kader AA, Stow JR, Colgan RJ, James DJ. Effect of down-regulation of ethylene biosynthesis on fruit flavor complex in apple fruit. Transgenic Res. 2004, 13(4):373-84, Latowski D, Kruk J, Strzałka K. Inhibition of zeaxanthin epoxidase activity by cadmium ions in higher plants. J Inorg Biochem. 2005, 99(10):2081-7), but the application of the methods requires optimization for lupine. Furthermore, the time planned to improve our Ms according to review reports is not enough to do such time-consuming analyses. Nevertheless, we are grateful for Reviewer suggestion. We focus on the role of phytohormones in the regulation of selected physiological processes, such as abscission. Therefore, we definitely try to introduce an efficient methodology of activity measurements for enzymes involved in ABA and ET biosynthesis. As Reviewer suggested, these data enriched our next papers and allow us to elucidate the mechanism that governs organ abscission.
Point 3: Figure 3C, 6A, 8A, Supplemental Figure 8C, 10C, 11A and 11B, change comma in the Y-axis to decimal point.
Response 3: It has been changed according to Reviewer suggestion.
Reviewer 2 Report
The text is interesting even if somewhat verbose but it is a paper of plant physiology that, at the moment, does not allow any consideration regarding the possible application of the results achieved at plant breeding level as it would seem instead to be inferred from what indicated in lines 54-59. The Authors should therefore modify the introduction by specifying the physiological interest of their research without attempting to give the idea that plant breeding approaches for yellow lupine in the Mediterranean area may arise from this work.
The species is quite important but, to my knowledge, Lupinus luteus presents three levels of ploidy, 2n = 46, 48 and 52; the Authors should clarify the level of ploidy of the materials used. Regarding the statistical approach, the work is poor, as for example the number of plants used for experimentation is not indicated. Furthermore, the difference between what is indicated on line 464 (biological replication) and what is eported on line 563 (technical replications) should be clarified. If what reported in the Figures are SEs based on two or three data, their statistical significance would be at least questionable. In general, the applied statistical approach should be much better explained in the Materials and Methods.
Other minor problems:
line 75: Arabidopsis thaliana
lines 91-101: transfer to Materials and Methods
lines 97-98: "... we observed disruption in the redox balance was shown to be broken ... .." the sentence should be reconsidered
line 133: P<0.05 and not 0,005
Figure 3: check what is reported in the figure for significance level (for example: what # does it mean?)
line 209: what does it mean in terms of correlation "... to similar correlation ...". Similar in what sense?
line 232: figure 8 and not 9
line 285: ---- also in Abies alba, ...
line 445: how many plants have been used? What accessions were they part of? What is the ploidy of the materials?
line 446: part of them (see what indicated for row 445)
line 447: Thauhan and Johnson not reported in biblio
lines 477 and 478: Krause et al., 1988 and Weng 2006 not reported in biblio
In general it would be appropriate, to favor the reading of the work, to present a Table indicating the meaning of the many acronyms used in the text.
Author Response
Response to Reviewer 2 Comments
Point 1: The text is interesting even if somewhat verbose but it is a paper of plant physiology that, at the moment, does not allow any consideration regarding the possible application of the results achieved at plant breeding level as it would seem instead to be inferred from what indicated in lines 54-59. The Authors should therefore modify the introduction by specifying the physiological interest of their research without attempting to give the idea that plant breeding approaches for yellow lupine in the Mediterranean area may arise from this work.
Response 1: Selected fragments of Introduction have been modified according to Reviewer suggestion.
Point 2:The species is quite important but, to my knowledge, Lupinus luteus presents three levels of ploidy, 2n = 46, 48 and 52; the Authors should clarify the level of ploidy of the materials used.
Response 2: Necessary information has been added to the Material and methods section (Taper variety 2n=52)
Point 3: Regarding the statistical approach, the work is poor, as for example the number of plants used for experimentation is not indicated. Furthermore, the difference between what is indicated on line 464 (biological replication) and what is eported on line 563 (technical replications) should be clarified. If what reported in the Figures are SEs based on two or three data, their statistical significance would be at least questionable. In general, the applied statistical approach should be much better explained in the Materials and Methods.
Response 3: AZ sections collected for analyses are quite small (approx. 2 mm fragments). Thus, material collected for experiments by us is very heterogeneous because we have to mix many small fragments from many different plants (and different inflorescences formed by each plant) to obtain sufficient amount of tissue for analyses. It should be noted that the amount of material (mixed AZs fragments from many plants) for analyses was different, e.g. for expression – 200 mg (approx. 50 AZ fragments), 500 mg for GC-MS analyses. We added some information regarding this issue in the Material and methods section.
Other minor problems:
Point 4:line 75: Arabidopsis thaliana
Response 4: Have been changed.
Point 5 lines 91-101: transfer to Materials and Methods
Response 5: We agree that this paragraph is too detailed and we changed it following Reviewer recommendation.
Point 6:lines 97-98: "... we observed disruption in the redox balance was shown to be broken ... .." the sentence should be reconsidered
Response 6: The actual version of Ms doesn’t contain this sentence.
Point 7:line 133: P<0.05 and not 0,005
Response 7: Have been changed according to Reviewer suggestion.
Point 8:Figure 3: check what is reported in the figure for significance level (for example: what # does it mean?)
Response 8: The figure has been improved. Proper symbols have been added.
Point 9: line 209: what does it mean in terms of correlation "... to similar correlation ...". Similar in what sense?
Response 9: Have been changed in the text.
Point 10: line 232: figure 8 and not 9
Response 10: Have been changed.
Point 11: line 285: ---- also in Abies alba, ...
Response 11: Have been changed according to Reviewer suggestion.
Point 12: line 445: how many plants have been used? What accessions were they part of? What is the ploidy of the materials?
Response 12: All significant information pointed out by Reviewer have been added to the Material and methods.
Point 13:line 446: part of them (see what indicated for row 445)
Response 13: The data have been added according to Reviewer suggestion.
Point 14:line 447: Thauhan and Johnson not reported in biblio
Response 14: The reference has been added.
Point 15:lines 477 and 478: Krause et al., 1988 and Weng 2006 not reported in biblio
Response 15: The references have been added.
Point 16:In general it would be appropriate, to favor the reading of the work, to present a Table indicating the meaning of the many acronyms used in the text.
Response 16: We have made a table with acronyms. It has been added to the Supplementary data.
Round 2
Reviewer 2 Report
The paper can be published in the actual form.